# Rituximab in Idiopathic Pulmonary Hemosiderosis in Children: A Novel and Less Toxic Treatment Option

**DOI:** 10.3390/ph15121549

**Published:** 2022-12-13

**Authors:** Suzanne W. J. Terheggen-Lagro, Eric G. Haarman, Niels W. Rutjes, J. Merlijn van den Berg, Dieneke Schonenberg-Meinema

**Affiliations:** 1Department of Pediatric Pulmonology and Allergy, Amsterdam University Medical Centres, University of Amsterdam, 1105 AZ Amsterdam, The Netherlands; 2Department of Pediatric Immunology, Rheumatology and Infectious Diseases, Amsterdam University Medical Centres, University of Amsterdam, 1105 AZ Amsterdam, The Netherlands

**Keywords:** pulmonary hemosiderosis, rituximab, pediatric, pulmonary bleeding

## Abstract

Idiopathic pulmonary hemosiderosis (IPH) is a rare, potentially life-threatening chronic disease. Steroids are the cornerstone of treatment, even though toxicity and side-effects are very common. Recently, rituximab (RTX) has been suggested as a treatment option, although evidence for its efficacy and long-term safety is lacking. We describe the disease course of two pediatric patients with IPH that were treated with RTX for over 4 years. Demographics, treatments, and clinical variables such as growth, infections, imaging follow-up by CT, and data from pulmonary function tests were retrospectively described. These are the first two cases described with a long-term follow-up of pediatric IPH patients treated with RTX. RTX was well-tolerated and prevented outbreaks of bleeding. In addition, RTX had a robust steroid-sparing effect resulting in the improvement of growth, pulmonary function, and CT abnormalities.

## 1. Introduction

Idiopathic pulmonary hemosiderosis (IPH) is a rare and potentially life-threatening disease that is mainly reported in children. This disease consists of recurrent bleeding in the lower respiratory tract, specifically in the alveoli, with an unknown pathophysiology. Various terms in the literature have been used for this phenomenon, such as diffuse alveolar hemorrhage (DAH) and acute idiopathic pulmonary hemorrhage in infants (AIPHI) [1]. DAH can form part of an underlying systemic (rheumatological) disease [2,3]. The term IPH is used as a diagnosis per exclusion. Since IPH is often unrecognized, there is an increased risk of mortality owing to diagnostic delay and, consequently, the delayed initiation of effective treatment. A recent overview showed that a diagnostic delay seems to occur more frequently in pediatric IPH, while in adults with IPH a higher prevalence of celiac disease antibodies has been described [4]. The golden standard for the diagnosis of IPH is histopathology from a lung biopsy with the detection of hemosiderin-laden macrophages without signs of vasculitis. This procedure is not without risk, especially with respect to the acute form of the disease. If a lung biopsy has not been performed, the diagnosis can be based on a combination of clinical symptoms (acute hemoptysis with/without fever), radiographic signs (multiple, focal, or diffuse alveolar shadowing on X-ray images and ground-glass opacities presented in a CT-scan), fluid analysis from broncho-alveolar lavage (BAL), and by excluding other causes [2]. In the long-term, IPH may be complicated by restrictive lung disease due to pulmonary fibrosis. 

By definition, the etiology of IPH is unknown but is assumed to be multifactorial, combining genetic predisposition, autoimmunity, allergic triggers, and other environmental factors [4,5]. Although the pathophysiology of IPH is not understood, immune-suppressive treatment is effective in the effected patients. Due to its favorable response to immunomodulatory drugs, Saha proposed to rename this condition to ‘immune-mediated pulmonary hemosiderosis’ [6]. By changing the name of this severe condition, with a high risk for mortality in the acute phase, clinicians might be persuaded to start immunosuppressive treatments earlier.

Systemic steroids are used in the acute setting but are also advised for maintenance therapy, despite their high drug toxicity [7]. In IPH, the majority of patients report symptoms within the first seven years of life [6]. Steroid toxicity is especially important in these young patients, as it can lead to growth restriction. Other side-effects include osteoporosis, hypertension, opportunistic infections, elevated blood glucose levels, muscular atrophy/weakness, acne, weight gain, stomach irritation, and mood changes. Recently, rituximab (RTX) has been suggested as a treatment option for IPH, but published data are limited to few case reports with limited follow-up [8,9]. No randomized studies are available. In addition, Saha et al. showed the first positive results of experimental treatment with mesenchymal stem cells in IPH [4].

This small case series describes the long-term follow-up of two pediatric patients with IPH that were treated with RTX for ≥4 years. The focus was to compare the pre- and post-rituximab treatment follow-up periods in terms of growth, pulmonary function, chest CT scans, and exacerbations. 

## 2. Results

In Table 1 the clinical characteristics, disease course, and treatments of the two patients included in this study are summarized. The first patient is now a 14-year-old boy (born in 2008) who first presented at the department of pediatric pulmonology at the age of 6 months with recurrent episodes of wheeze and lower airway infections. He was born at a gestational age of 35 weeks and 2 days with a birth weight of 2580 g. Immediately after birth, he was treated with antibiotics and supplemental oxygen under suspicion of a neonatal pneumonia. Since then, he has had recurrent episodes of dyspnea with persistent crackles and intermittent wheeze on auscultation. 

At the age of 6 months, a bronchoscopy was performed: the anatomy was normal, BAL-fluid showed non-specific inflammation, no lipid-laden macrophages were identified, and bacterial cultures were negative. A respiratory epithelial biopsy showed normal ciliary motility and ultrastructure. Extensive immunological screening of the patient’s blood did not reveal any abnormalities or signs of allergies. His complaints were considered suitable for viral wheeze and he was treated with inhaled corticosteroids, beta 2 reliever therapy, and ipratropiumbromide. He suffered from recurrent pulmonary exacerbations consisting of tachy- and dyspnea, crackles, wheeze, and intermittent abnormalities in chest X-rays suggestive of consolidation, for which he was treated with prednisolone and antibiotics. 

At the age of 4 years, he suddenly presented with severe anemia, a patchy consolidation in a chest X-ray, and pulmonary hemorrhage was suspected. After this episode, bronchoscopy and BAL were repeated showing hemosiderin laden macrophages. Similarly, hemosiderin-laden macrophages were identified in a lung biopsy without signs of capillaritis. An extensive blood work-up (auto-antibodies, celiac disease, and cow’s milk allergy) and cardiac screening did not show any abnormalities. Based on these test results, the patient was diagnosed with idiopathic pulmonary hemosiderosis (IPH), and treatment consisting of systemic steroids, mycophenolate mofetil, and antibiotics (azithromycin) in addition to inhaled corticosteroids and hydroxychloroquine was started. In the following years, he experienced multiple exacerbations for which frequent systemic steroids via pulse therapy were needed. This effectively stopped the pulmonary bleeding but did not prevent a further decline in lung function. In addition, the patient experienced the following severe side effects of the systemic and inhaled steroid use: the restriction of linear growth, weight gain, muscle weakness, infectious problems (varicella and fungal infection), behavioral problems, and adrenal insufficiency. 

At the age of 9 years, RTX treatment was started. In the following 5 years, he experienced only two exacerbations for which he needed treatment with methylprednisolone. Maintenance treatment with oral prednisolone was tapered and stopped at the age of 10. Adrenal insufficiency still remains but maintenance treatment with hydrocortisone has now been gradually lowered. Chest CT scans showed fewer ground glass opacifications but an increase in interlobular septal thickening over time in accordance with the previous years of severe exacerbations and instable disease (Figure 1). The patient’s lung function, which had been rapidly declining, now shows a gradual increase in forced vital capacity (Figure 2). In addition, the fatigue that severally hampered his daily activities has improved considerably. Due to the RTX treatment, his IgG levels declined just below the normal limit (4.9 g/L). After recurrent viral respiratory infections resulting in two milder IPH exacerbations (the first shortly after the start of RTX treatment, while for the second no admission was needed), suppletion with intravenous immunoglobulins (IVIG) was started. Since then, no outbreaks of IPH have occurred. His height is catching up after steroid administration was stopped following stunted growth for many years (Figure 3). Due to new diagnostic possibilities, he was screened in 2019 for interferonopathy, which showed a mild elevated interferon type 1 signature. Nevertheless, genetic testing showed no abnormalities, specifically, no signs of the then newly described COPA syndrome (coatomer protein complex subunit alpha), for which JAK-inhibition might have been an effective treatment. In conclusion, RTX treatment is well-tolerated with mild hypogammaglobulinemia, for which he receives supplemental immunoglobulin therapy every 6 weeks. 

The second patient is a now 16-year-old girl (born in 2006) who presented to another hospital at the age of 4 years with anemia and pneumonia. She was treated with antibiotics and diagnosed with iron deficiency for which supplemental iron treatment was started. After initial presentation, she had recurrent episodes of anemia and was prescribed supplemental iron treatment several times. At the age of 5 years, she presented to our hospital with appendicitis for which she needed surgery. A further work-up of her iron deficiency and anemia showed no clear explanation. At the age of 6 years, she presented with a third episode of pneumonia and anemia, for which a diagnosis of pulmonary hemorrhage was suspected based on patchy infiltrates on a chest X-ray and the combination of her history of recurrent anemia and pneumonia. A further work-up consisted of bronchoscopy and BAL showing hemosiderin-laden macrophages and a lung biopsy that showed iron-loaded macrophages without signs of capillaritis. Furthermore, an extensive blood work-up (auto-antibodies, celiac disease, cow’s milk allergy, and genetic testing) and cardiac screening were performed, which did not show any abnormalities. Based on these test results, the patient was diagnosed with IPH and treatment was started. Her treatment regimen consisted of systemic steroid maintenance therapy and antibiotic prophylaxis. She experienced one exacerbation every 2 years, while the side effects from systemic steroids were behavioral changes and excessive weight gain, which led to low compliance. One year later, she presented at our hospital with respiratory failure due to diffuse alveolar hemorrhage that required ventilatory support and methylprednisolone pulse therapy. She recovered within a week and restarted systemic steroid maintenance therapy and hydroxychloroquine and mycophenolate mofetil treatment. In the following 3 years she experienced five exacerbations of pulmonary hemorrhage. She showed a progression of ground glass opacifications on her chest CT-scan in accordance with active disease. She also experienced side effects of the systemic steroids as mentioned before and reported lower adherence to the oral medication during adolescence. Therefore, RTX treatment was started at the age of 12 years. In the 4 years thereafter, she only had one exacerbation during RTX treatment due to a delay in treatment during the summer holidays. She tolerates ongoing RTX treatment very well and without side effects and chest CT scans show improvement with fewer ground glass opacifications (Figure 4). She also shows a gradual increase in total lung capacity and forced vital capacity on lung function measurements (Figure 5). 

## 3. Discussion

This is the first case series that describes in detail the long-term effects of RTX treatment in children with IPH. These two cases show that RTX can effectively replace systemic steroid treatment in IPH, leading to fewer exacerbations and lowering corticosteroid-related side-effects. During their long-term follow-ups, both patients presented stable disease and an improvement in lung function. The second patient even showed an improvement in structural abnormalities on a chest CT scan. 

Historically, systemic steroids have been the cornerstone of treatment of IPH. Steroid-sparing agents such as azathioprine and hydroxychloroquine have been used in IPH with limited success. RTX has only been reported in incidental cases with refractory disease. Our patients both suffered from ongoing exacerbations, steroid dependence, and major steroid-induced side-effects, despite additional hydroxychloroquine and mycophenolate mofetil maintenance therapies. These exacerbations resulted in progressive damage of their lungs, as shown on chest CT scans, and an increase in steroid-related toxicity. Both patients were diagnosed at a very young age and were exposed to chronic steroid treatment for years during a critical phase of growth and development. The toxicity of (long-term) steroid use is a well-known phenomenon, but despite this toxicity, long-term treatment with steroids is still advised for IPH patients. In contrast, in treatment protocols for children and adults with systemic lupus erythematosus (SLE), there is currently a shift towards a more proactive use of steroid-sparing regimens [10,11]. In our two patients, corticosteroid-related toxicity was evidently lowered by starting RTX treatment.

Both patients reported adherence difficulties with respect to their maintenance treatment regimens during their teenage years. This is a well-known phenomenon and a great challenge to the medical teams and parents caring for teenagers with chronic diseases. Rituximab is better tolerated, and adherence is guaranteed by intravenous administration, which is only carried out once every 6 months.

The intravenous administration of RTX is effective in the treatment of systemic autoimmune diseases such as vasculitis. The anti-inflammatory effect of RTX is caused by B-cell depletion leading to the decreased production of auto-antibodies. In IPH, there is no sign that autoimmunity has a role in its pathophysiology; therefore, the positive effect of auto-inflammatory treatment in IPH is not fully understood. Some patients show signs of interferonopathy caused by genetic mutations leading to problems in the innate immune system [12,13]. It has also been suggested that during the disease course the autoimmune serology can change over time with later positivity for ANCA auto-antibodies and other systemic ANCA-associated symptoms [9,14]. ANCA-positivity (without other signs) might also be a non-specific finding and reflect the risk for autoimmunity in a patient. In one of the publications, a child with presumed IPH since the age of 4 was successfully treated with RTX, but later turned out to be MPO-ANCA-positive at 16 years of age [9]. In ANCA-positive vasculitis, rituximab is a well-described and effective treatment. In our patients, ANCA-screening was repeatedly negative, as well as in the years following the diagnosis.

In our patients, there were some but mild signs of active bleeding in the first year after starting RTX treatment, but during the first year, both patients could stop frequent steroid (pulse) therapy and exacerbations were only mild or sporadic. This suggests that when treatment with RTX is started in IPH, some time is needed before an optimal therapeutic effect is achieved. 

We realize that the evidence from this small case series is limited. However, in these patients, the steroids’ drug toxicity was severe, and these toxic effects could evidently be lowered in our patients after starting RTX treatment with a steroid-sparing effect. In case of relapse and/or steroid dependence, we suggest RTX as a maintenance therapy in pediatric IPH patients.

## 4. Materials and Methods

This case series describes two pediatric IPH patients that were treated in a tertiary center at Emma Children’s Hospital/Amsterdam UMC between 2011 and 2022. In our hospital, these patients were treated by the pediatric pulmonologists in collaboration with the pediatric rheumatologists/immunologists. Demographics; type and start/stop dates of different treatments; clinical variables such as growth, infections, and imaging follow-up by chest X-ray and CT; and data from pulmonary function tests were retrospectively collected from patients’ charts. Written consent from patients and their parents has been obtained.

## 5. Conclusions

This is the first report that describes the long-term follow-up of pediatric IPH patients treated with RTX. RTX appears to be an effective, safe, and steroid-sparing treatment modality for the long-term treatment of pediatric IPH, thereby diminishing the need for systemic steroid treatment.

## Figures and Tables

**Figure 1 pharmaceuticals-15-01549-f001:**
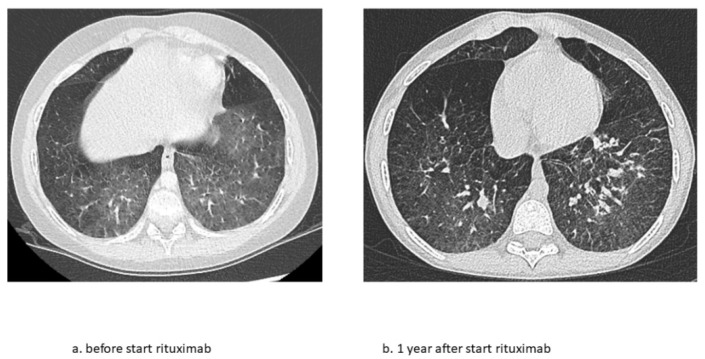
Chest CT scan images before and 1 year after start of RTX treatment during clinically stable disease phase. (**a**) Shows ground glass opacifications as main abnormality and (**b**) less well-defined ground glass but more interlobular septal thickening after 1 year of RTX.

**Figure 2 pharmaceuticals-15-01549-f002:**
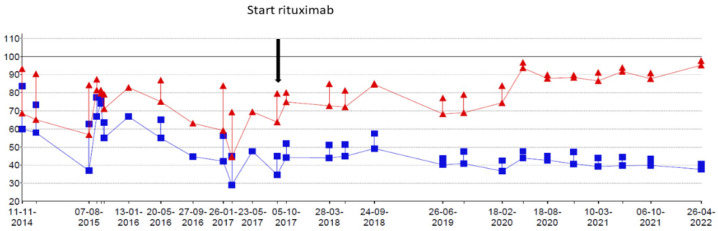
Lung function over time (forced expiratory volume in 1 s (FEV1) in blue and forced vital capacity (FVC) in red expressed as percent predicted of normal. FVC shows a gradual increase while FEV1 remains stable.

**Figure 3 pharmaceuticals-15-01549-f003:**
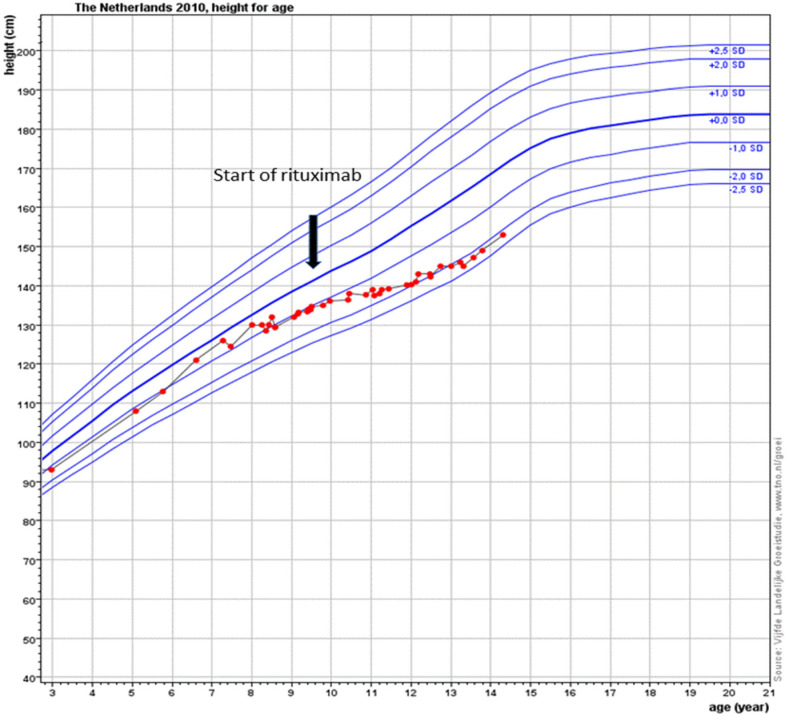
Red dots and red line reflect the different height measurements over time for this patient compared to the reference height growth lines SD lines) for boys in the Netherlands. Growth in height over years expressed by the red dots. It is clearly shown that the systemic steroids caused a stunting of height with a gradual improvement in height growth velocity after starting rituximab.

**Figure 4 pharmaceuticals-15-01549-f004:**
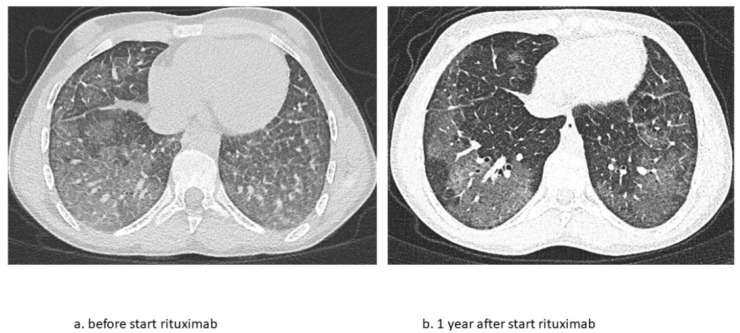
Chest CT scan images before and 1 year after start of RTX treatment during clinically stable disease phase. (**a**) Ground glass opacification as main abnormality and (**b**) fewer ground glass opacifications after 1 year of RTX.

**Figure 5 pharmaceuticals-15-01549-f005:**
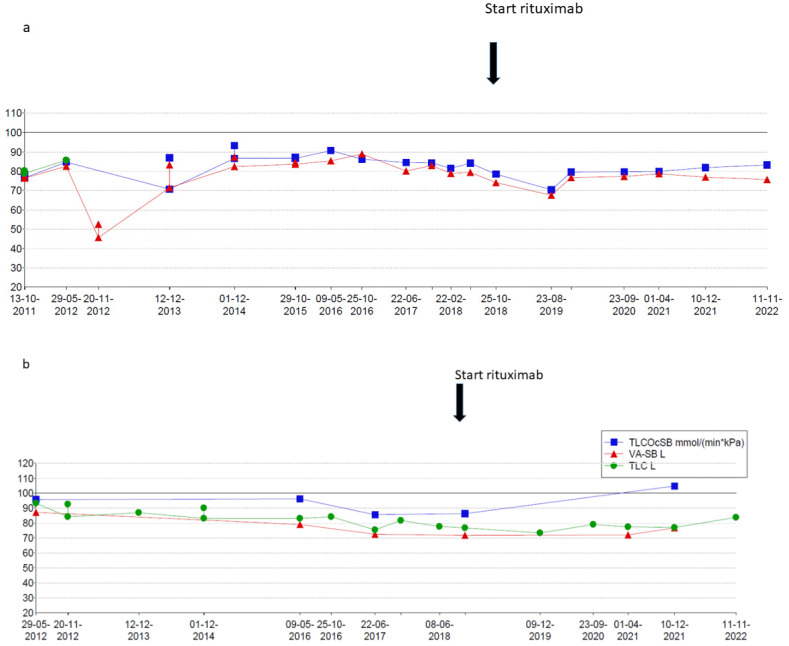
(**a**) Lung function over time (forced expiratory volume in 1 s (FEV1) in blue and forced vital capacity (FVC) in red expressed as percent predicted of normal. (**b**) Total lung capacity (TLC) in green measured by body plethysmography and total lung capacity measured by CO-dilution technique (TLCO-SB) in green and alveolar volume (VA-SB) in red. All values are expressed as percentage predicted of normal.

**Table 1 pharmaceuticals-15-01549-t001:** Patient characteristics, disease course, and treatment of 2 patients with idiopathic pulmonary hemorrhage. RTX, rituximab; MPNS, methylprednisolone; IVIG, intravenous immunoglobuline.

Patient Characteristics	Patient 1	Patient 2
Age at onset	Shortly after birth?	4 years
Age at diagnosis	4 years	6 years
Age at start RTX	9 years	12 years
No. of prednisone episodes needed (pre-RTX)	2–6 times/year	1–2 times/year
Stop prednisolone (in months after start RTX)	15	9
MPNS pulse after RTX (until follow-up end of 2022)	2	1
Steroid-related toxicity	Inhibition of linear growth, weight gain, muscle weakness, behavioral changes, adrenal insufficiency	Behavioral changes, weight gain
Rituximab complications	Hypogammaglobulinemia: IVIG (400 mg/kg every 6 weeks)	-
Interferon signature (IFN)	IFN type 1: score 10 (mildly elevated >9, under treatment 2019)	IFN type 1: <1 (normal, under treatment in 2019)
Genetic testing	Panel for primary immunodeficiency: negative (2018) Trio-Whole-Exome Sequencing with parents: negative (2022)	Awaiting test results of Whole Exome Sequencing (2022)

## Data Availability

Data is contained within the article.

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
