# Peer review of "Rituximab in Idiopathic Pulmonary Hemosiderosis in Children: A Novel and Less Toxic Treatment Option"

_pharmaceuticals, 2022, doi:10.3390/ph15121549_

Round 1

Reviewer 1 Report

I thank the authors for reporting these interesting and important cases. These cases will certainly help clinicians manage patients with IPH throughout the world. The authors have elegantly presented the cases and discussed important topics in the management of IPH. I have a few minor suggestions.

1.       Introduction: Saha et al. have recently proposed a new hypothesis for the pathogenesis of IPH. I would include that in the section where you discuss the pathogenesis (2nd paragraph). This hypothesis explains the role of immunosuppression for patients with IPH. It will fit nicely before you make the statement about the efficacy of immunosuppressants in IPH. doi.org/10.1007/s00408-022-00523-4

2.       Case presentations look good.

3.       Discussion: looks great. I agree with the authors completely that IPH is often sub-optimally managed compared to rheumatologic diseases and aggressive management is necessary to prevent complications or at least delay disease progression.

The INF 1 signature is an interesting concept and further investigation is certainly necessary.

Other:

A recent review summarized the latest developments in the field of IPH, including experimental treatments. I would recommend the authors cite that paper for the readership. doi:10.1002/ppul.26230

Author Response

dear reviewer, thank you so much for your compliments and useful feedback and comments.

For comment 1: we agree and have transferred this sentence from the discussion part to second paragraph of the introduction. 

Other: thank you for this comment, this recent published paper will be incooperated in our manuscript in the introduction and in the references.

Reviewer 2 Report

The authors describe two pediatric patients with Idiopathic pulmonary hemosiderosis (IPH) and present the results of the disease course and long-time patients' treatment with Rituximab (RTX). It reviewed a large number of clinical and therapeutic data. The authors emphasize that RTX was well tolerated and prevented flares of bleeding, and simultaneously RTX treatment leads to a robust steroid-sparing effect with the improvement of growth, pulmonary function, and CT abnormalities. 

The authors used a large number of specific methods that fully correlate with the purpose, result, and discussion. The study is systematic, and data analyses and results are interesting and well-presented. The manuscript needed minor spelling corrections.

Author Response

dear reviewer, thank you so much for your compliments and suggested minor revisions. We will thoroughly walk through the full manuscript for correction of faults in spelling.